# Tell Me Where to Go: An Experiment in Spreading Visitor Flows in The Netherlands

**DOI:** 10.3390/ijerph20085441

**Published:** 2023-04-07

**Authors:** Ondrej Mitas, Rajneesh Badal, Maud Verhoeven, Koen Verstraten, Liselotte de Graaf, Helena Mitasova, Wendy Weijdema, Jeroen Klijs

**Affiliations:** 1Academy for Tourism, Breda University of Applied Sciences, 4817 JS Breda, The Netherlands; 2Zoey, 1066 JS Amsterdam, The Netherlands; 3Center for Geospatial Analytics, College of Natural Resources, North Carolina State University, Raleigh, NC 27695, USA; 4Marketing Oost, 8041 BL Zwolle, The Netherlands

**Keywords:** emotion, experience, tourism, tourist flows, spatial behavior, recommender systems

## Abstract

Cities attracting large numbers of tourists increasingly face crowding and public resistance to tourism growth. As a result, governments strive to spread tourists from the best-known attractions to less-visited locations to improve both residents’ and tourists’ quality of life. Evidence of success and best practices herein is largely anecdotal, and the effects on tourist experience are also unknown. Thus, we undertook a randomized 2 × 2 experiment in the province of Overijssel (The Netherlands), wherein tourists staying at vacation parks near small and mid-sized cities were exposed to information which emphasized attractions in either heavily visited or less-visited areas. Participants were also assigned to receive the information in either a passive or a conversational form. Location and daily emotion, as well as experience evaluation on the last day of the vacation, were recorded via mobile platforms. We found that tourists receiving information on attractions in less-visited areas engaged in significantly more movements around these attractions, and significantly less around heavily visited areas. The conversational form of information delivery was more positively evaluated than information delivered passively. Furthermore, vacation experience emotions and evaluations were largely unaffected. Thus, it is clearly possible to direct tourists to less-crowded locations without negatively affecting their vacation experiences.

## 1. Introduction

Cities that attract large numbers of tourists increasingly face crowding and public resistance to tourism growth. This phenomenon is best-documented, though not limited to, cities, and has been termed *overtourism* [1]. Overtourism is detrimental to tourists’ experiences as well as to local residents’ quality of life. As a result, city governments strive to spread tourists from the best-known attractions to less-visited urban and peripheral locations.

An alleged mechanism of tourist spatial behavior is information that tourists consume. This information includes guidebooks, review sites, and social media. It is believed that the most-used information sources tend to echo one another, steering tourists to a relatively small number of attractions. At the same time, other places where tourist could have enjoyable experiences, remain undiscovered and suffer, to some degree, from undertourism. Important in existing visions for resetting tourism after the pandemic is bringing these under-/and over-visited places more into balance. For example, the Destination Management Organization (DMO) of The Netherlands, called The Netherlands Board of Tourism and Conventions (NBTC), wrote in 2019, thus based on pre-pandemic tourism levels, that “Heading up to 2030, we expect a 50 percent increase in the number of international tourists. This requires a new approach that prioritises the common interests of visitors, companies, and residents. The goal is for every Dutch person to benefit from tourism. Five priorities are central for achieving this ambition, [including to] put more cities and regions on the map as attractive destinations” [2].

Therefore, in the present study, we aimed to test an intervention to inform tourists about attractions in less-visited areas, in contrast to attractions in heavily visited areas. How information provided by DMOs affects tourist flows is generally unknown, and no studies exist as far as we could find that have manipulated experimentally and then tracked spatial behavior by tracking tourists’ locations. We carried this intervention out over two distinct information channels to uncover how information *as well as* how it is delivered might influence spatial behavior. One of these was a conversational recommender system, an under-researched method for tourist information delivery that has been touted as promising [3] yet rarely tested. We tested a conversational recommender system, which is an information system that offers highly personalized information in a chat-like environment, against a passive recommender system, which displays tourism information on a destination map with no personalization. Besides measuring spatial behavior, our study also addressed tourists’ experiences, to assess the impacts of intervening on information on local communities as well as on tourists themselves. The ultimate goal of this inquiry was to assess informational tools at DMOs’ disposal in their effect on tourism flows, so that information provision choices can be made with both tourists and local residents in mind.

## 2. Literature Review

### 2.1. Tourism Flows and Overtourism

Knowledge about the impacts of tourism flows on host resident communities is nearly as old as academic research of tourism. In fact, publications such as Turner and Ash’s Golden Hordes [4] and Doxey’s instrument of local resident irritation, termed the Irridex [5] actually predate most early studies of vacation experience. The Destination Life Cycle model posited that these impacts unfold over time, and eventually lead to declines of destinations [6]. The negative sides of tourism impacts were documented and explained by a variety of theories in thousands of publications. Later, the dramatic expansion of low-cost airline flights in the 1990s had a role in accelerating and localizing such impacts [7] More recently, after global economic recovery from the 2008 financial crisis, perceived negative tourism impacts on host residents exploded into outright conflict, most prominently in European cultural tourism hubs such as Venice and Amsterdam [8].

At the same time, the term “overtourism” appeared increasingly often in academic journals [1,9,10] as well as popular media. While academic definitions of the term barely differ from those in the Irridex and Life Cycle, the conversation around overtourism also implied that two new boundaries had been approached. First, visitor levels were so high on some streets that it was not physically safe or possible for these spaces to accommodate more visitors. Second, tourism impacts on local residents were not merely an unfortunate consequence of tourism development. Instead tourism impacts have been increasingly viewed by local residents as a threat to livability and life itself [1], and thus worthy of political or even violent resistance against tourism growth. However, this phenomenon has been spatially limited. For example, the average resident of Amsterdam still felt positive about levels of tourist flows in 2017 [10], although quality of life for residents in a few specific, heavily visited neighborhoods had declined alongside a rise in tourism-related disruptions [11].

Previous research thus demonstrates that economic motivations to increase tourist flows and societal motivations to reduce or redirect them have long been in conflict. Unfortunately, academic research on overtourism has mostly extended the tradition in tourism impacts research of precisely describing and explaining problems rather than proposing and testing solutions. Most proposed solutions from tourism scholars, such as community-based creative tourism, have merely niche appeal [12] and require high-cost, high-impact travel over long distances that prices out most potential tourists and is associated with high CO_2_ emissions. Meanwhile, mass demand among European and American source markets for heavily visited destinations has rebounded dramatically after pandemic-related restrictions of 2021 and 2022 were lifted. Thus, the struggle to manage tourism flows and to ease tourist-resident conflict continues.

Potential solutions have emerged from both destination management as well as technological arenas. As an example of destination management, Venice has decided to charge an admission fee to enter the city. Recent visual scenario-based research of crowding in Amsterdam showed wide-ranging support for an entry fee to that city [8]. Another category of potential solutions pertains to how tourists are informed. There is remarkably little academic research on the connection between destination information tourists consume and their behavior on-site. Research about tourist decision formation instead focuses heavily on destination choice and other pre-travel decisions [13,14,15,16]. Yet, it is widely acknowledged that tourist behavior on-site has always been formed by information such as guidebooks and postcards [17,18], though it is not understood precisely how. Therefore, changing information about activities and attractions at the destination appears to be a promising but severely under-researched method for redirecting tourist flows.

The pandemic restrictions of 2020 and 2021 highlighted the importance of information as a possible method of altering tourist flows. While tourism demand in urban environments declined, social distancing meant that any existing visitor flows needed significantly more physical space. Once again, solutions emerged from both destination management as well as technological arenas. Several destinations such as Texel, an island in The Netherlands, created apps which not only provided information about possible attractions, but also indicated how busy they were, with the goal of spreading tourist flows away from the busiest locations. Google Maps has also provided this functionality across an increasing variety of location types (trains, cafes, public spaces). Furthermore, new communication channels such as short-format video and chatbots have been adopted by destinations. Among the most promising of these have been conversational recommender systems, a technical innovation which has been applied to tourist information [19,20], though not researched in the context of managing tourist flows.

### 2.2. Conversational Recommender Systems

A conversational recommender system occupies the space between a completely personal conversation with a human being and a completely automated conversation with a chatbot. While personal conversations have long been a staple of DMOs via brick-and-mortar offices, phone lines, and direct messaging on social media, they comprise an expensive information method. Chatbots, on the other hand, quickly fail and frustrate recipients when questions become intricate or unusual. A recommender system is defined by [21] (p. 56) as “a system that uses personal data or preferences as input data, after system processing and cross-check, to provide recommendation results to the appropriate recipients.” Research has shown that there are four core dimensions which give value to a tourist recommender system: quality information, easy access on mobile screens, fast and interactive communication, and personalization [3,22,23]. The last dimension, personalization, has been an area of vigorous innovation over the past decade. The main technical approaches to personalization in the past included filtering information based on content provided by end users or users’ personal/demographic characteristics. Such filtering is a rather blunt tool and has led to artificial and often unhelpful conversations.

Recent improvements in recommender systems have come from artificial intelligence. Recommender systems become more complete, accurate and user friendly [24]. In the context of recommender systems, artificial intelligence is defined as “a system’s ability to interpret external data correctly, to learn from such data, and to use those learnings to achieve specific goals and tasks through flexible adaptation” [25]. In the present study, we test the delivery of attraction information to tourists using the conversational recommender system Zoey. Zoey uses artificial intelligence to learn from the contact moments with the customer and to build a user profile which is more personalized than a one based on filtering. Furthermore, Zoey is not fully automated, but also deploys human employees to select and enhance responses before they are passed on to users. The result is an implementation called Travel with Zoey which mobilizes the convenience of a mobile chat function alongside the personalization level of a visit to a brick-and-mortar tourist information center. The mobile chat function is integrated in an existing mobile chat application such as WhatsApp.

The content from which Travel with Zoey generates responses is stored in a digital catalogue. Each attraction in this catalogue is tagged with identifiers such as ‘restaurant’, ‘vegetarian’, and ‘family’, facilitating an optimal match between tourist and content. An unsolicited message with attraction suggestions is sent every morning, but Travel with Zoey also responds to request by the customers themselves. Behind the scenes, the delivery of the content is mostly but not fully automated, so that human employees of Travel with Zoey are involved in making each interaction more personalized than a chatbot could. There is an emphasis on empathy, wherein Travel with Zoey aims to make customers feel heard and attended to. Finally, any experiences at attractions that customers talk about in their conversation with Zoey are taken by Travel with Zoey employees as input for optimizing future recommendations. As for other recommender systems, the goal is to not only provide a quality experience in consuming destination information, but to improve the tourist experience as a whole [26,27].

### 2.3. Tourist Experience

While a comprehensive review of tourist experience is beyond the scope of this article see [28,29], we touch on this construct from two relevant angles: to assert a contemporary psychological view on experience and how it is measured, and to relate experience to tourists’ attraction choices and resulting spatial behavior. In conceptualizing experience, we follow current knowledge in cognitive psychology and neuroscience to define experience as a process wherein the continuous sense of experiencing is selectively recalled and evaluated, sometimes termed “the experiencing self” and “the remembering self” [30]. The mechanism of selecting episodes as they are experienced for encoding into memory is widely understood to be emotion [31]. In other words, when something triggers sufficiently intense emotion in an individual, that individual will remember it. It follows that measuring an experience as it unfolds is best done by measuring participants’ emotions, ideally as continuously and unobtrusively as possible [31,32]. During a heterogenous experience such as tourism, using a daily emotion diary strikes a balance between recall error and participant burden [33]. In measuring recalled tourist experiences, in contrast, there is often a need to evaluate quality––to what extent the experience-as-a-whole was a good one, and worth the resources invested. Intent to recommend and single-item grading items are used as ‘gold standards’ for this sort of measurement [34,35,36], although longer scales to measure dimensions such as memorability [37] and impact on personal life [38] exist.

There is remarkably little known about how *locations where* tourists go affects their experiences, yet this is a key concern for DMOs aiming to redirect or disperse tourist flows partly away from the most popular attractions. Existing research has assessed connections between the continuous sense of experiencing a vacation and how it is recalled and evaluated [32,34,39] as well as internal [40] and social [41] determinants of daily experiences during tourism. There has been no systematic effort to link either continuous experiencing or recalled experience to spatial dimensions of tourist experience. It is worth noting that Kirillova and colleagues have linked destination beauty, which is likely to vary by location, to satisfaction [42] and the restorative qualities of tourist experiences [43]. Thus, it is likely that tourist experience, operationalized as daily emotion and post-experience evaluation, varies by spatial behavior. A small handful of studies used innovative physiological recording techniques to demonstrate variation in emotion across attractions in urban spaces of, for example, Jerusalem [44], Nuenen (The Netherlands) [32], and Lleida (Spain) [45]. These studies only looked at a single episode comprising a short walk through the urban space, not a tourist experience in its entirety, which typically comprises multiple transport modes, cities, and different mixes of attractions visited on different days.

### 2.4. Our Study—Justification and Approach

While the technical and user experience value of a recommender system such as Zoey have been well documented in previous research, the application of this technology to overtourism issues has not been studied. If information interventions from recommender systems or more conventional sources actually causes tourists to travel elsewhere, it is likely to affect their emotional and evaluated experiences, but this assumption remains untested. Tourists’ emotions drive their evaluations, however, which in turn drive important destination outcomes such as behavioral intentions [34]. Thus, it is important to understand how tourists’ spatial behavior affects their experiences. There are several specific knowledge gaps resulting from this lack of research. It is not known to what extent tourists’ on-site attraction choices, and therefore their spatial behavior, is affected *either* by the information they receive *or* how they receive it. It is not known whether manipulating such information is effective in redirecting tourism flows to less-visited areas. Thus, DMOs do not currently understand if intervening on information is worth the investment. Furthermore, it is not known whether a conversational recommender system is more or less effective in redirecting tourism flows than existing default information systems, and thus merits potential technical challenges in implementation. Finally, it is not known whether tourists who are accordingly redirected experience and evaluate their vacations differently as a result, which has consequences for them personally, as well as for the destinations they visit.

In this project we utilized Travel with Zoey as a conversational recommender system to deliver destination information to a group of visitors to Overijssel, a province in the east of The Netherlands. We compared its effectiveness to that of a default, passive map-based mobile application. Furthermore, we investigated the effectiveness of prioritizing attractions based on the policy of the DMO to direct visitors to certain places while reducing the pressure on others. We were guided by the following question:

Does a conversational recommender system, as compared to passive map-based mobile applications, spatially direct tourists to less-visited areas, and how are their vacation experiences and evaluations changed as a result?

More specifically, we address the following sub-questions in our study:

To what extent does tourists receiving different attraction information (attractions in heavily-visited vs. less-visited areas), from different sources (conversational vs. passive),

differ in their spatial movements?visit different types of attractions?experience different self-reported emotions day-to-day?visit different proportions of urban and rural destinations?evaluate their experiences differently?

## 3. Materials and Methods

### 3.1. Study Design

As we wanted to measure the specific and direct effects of conversational recommender systems and the information therein on vacation behaviors and experiences, we used a an experimental design with random assignment to intervention conditions, which is the only research design that supports conclusions about an intervention *causing* a particular outcome [46]. Two independent variables were manipulated: whether participants were invited to use a conversational recommender system (the *conversational* conditions) or a conventional, passive, non-conversational map-based application (the *passive* conditions), and whether they received information about attractions in heavily visited areas (the *popularity-driven* conditions) or in less-visited areas (the *policy-driven* conditions). Thus, participants were randomly assigned to receive destination information in one of four conditions:Popularity-driven information via a conventional passive map app;Policy-driven information via a conventional passive map app;Popularity-driven information via a personalized conversation on WhatsApp;Policy-driven information via a personalized conversation over WhatsApp;

Given that DMOs have traditionally informed tourists about attractions based on feedback about attraction popularity, and that numerous larger DMOs boast a passive map-based app as one of their information channels, the popularity-driven passive condition approximates a control condition in our design. It is challenging to study interventions on tourism experiences in the field with meaningful control conditions, as the experience must remain intact. As tourist experiences without information interventions, such as the use of a conversational recommender or purely policy-driven attraction information, still often feature passive map-based apps with destination information, we argue that this condition comprises a plausible control condition.

Information about attractions in the region were based on a database of approximately 400 attractions provided by Marketing Oost, the DMO for the province of Overijssel. Marketing Oost categorized the attractions based on priority. The best attractions were to be highlighted as having priority, while other attractions were displayed to participants without distinction. Which attractions were considered ‘best’, however, differed in location based on experimental condition. A different, mutually exclusive set of attractions was given priority in the popularity-driven conditions and in the policy-driven conditions. For popularity-driven participants, attractions judged by Marketing Oost as relatively higher quality, and located in the *most heavily* visited areas of Overijssel, were given priority. Conversely, for participants in the policy-driven condition, attractions judged by Marketing Oost as relatively higher quality, but located in *less-visited areas*, were highlighted as priority. On the passive map-based apps, priority attractions were in color, whereas non-priority attractions were grey. In the conversational recommender system, priority attractions were recommended to participants in any instance when multiple recommendations were possible. In both conversational and passive conditions, the same database of attraction information was available. However, calling up these attractions was based on responding to and asking for information in the conversational condition (Figure 1a), and moving around on the displayed map in the passive condition (Figure 1b).

As a region benefiting from tourism, but also facing intense tourist flows in a handful of heavily visited areas, Overijssel is an ideal context for the present study. This province of 1.16 million inhabitants is located in the northeast of The Netherlands, and borders Germany to the east and other Dutch provinces to the north, south, and west. The province covers a total of 342,273 hectares, of which 6773 (1.98%) are covered by recreational areas and 34,474 (10.07%) by forest. While existing discourse about overtourism largely concerns cities, especially large, internationally famous historic city centers such as Venice or Amsterdam, the situation is somewhat inverse in Overijssel. While there are several Hanseatic historic city centers which attract tourists, such as Zwolle (2021 population 129,840) and Deventer (2021 population 101,236), these cities are considered relatively less-visited areas by Marketing Oost for the purpose of the present study. In contrast, the town of Ootmarsum (2021 population 4460), the village of Giethoorn (2021 population 2805), and several natural and agricultural landscapes such as the Vecht valley and the Sallandse Heuvelrug National Park comprise the heavily visited areas of Overijssel. Thus, the policy to spread tourists away from heavily visited areas in Overijssel corresponds with the goal to increase visits to urban areas. In contrast, a region such as Amsterdam might indicate fewer tourists in urban areas as a policy goal to reduce crowding.

The experiment featured a during- and post-intervention measurement design. Participants were asked to fill in an intake questionnaire, including demographics and informed consent, approximately one month before their vacation. Location and emotions were measured daily during the vacation. Finally, on the last day of their vacation, participants were asked to evaluate the information source to which they were assigned (passive app or conversational recommender) and their vacation as a whole.

### 3.2. Sample

We used an availability sampling approach during two waves of data collection in May 2021 and late July/early August 2021. For field research in tourism contexts, availability samples being used as probability samples is practically impossible due to the lack of a sampling frame. Even if a sampling frame such as all visitors entering a region were practically possible, not only do tourist behaviors and experiences differ season to season, but also year to year. That makes a true probability sample of tourist experiences not only practically, but also conceptually unattainable.

Participants were approached based on the criterion of booking a vacation at one of 10 (May) or 8 (August) vacation parks that chose to cooperate with the project. These parks are commercial operations which offer varying mixes of camping and bungalow accommodations, occasionally with dining, basic shopping, and activity facilities. A lottery to win back the cost of one’s vacation accommodation was offered as an incentive. If would-be visitors to these parks agreed to participate, we sent them an intake questionnaire including a statement of informed consent. The intake survey assessed demographics. Five days ahead of the beginning of their vacation, they received instructions for connecting with their assigned information source, comprising of either adding a WhatsApp contact with Travel with Zoey (conversational conditions), or downloading the appropriate mobile application, either Nienke’s Tips (passive popularity-driven condition) or Saar’s Tips (passive policy-driven condition). They were also instructed to download and install a GPS tracking application, Sesamo, which additionally notified them every evening to fill out a daily self-response questionnaire. Besides measuring self-reported emotions, this questionnaire asked them if the current day was the last day of their vacation. If so, experience evaluation measures were displayed as well. The final sample size varied by the data sources used in each analysis; 268 participants filled in at least one daily questionnaire, while 155 of these provided GPS data and 132 responded to the last-day questionnaire including experience evaluations.

### 3.3. Measures

#### 3.3.1. Location

We tracked location using GPS via the Mobidot mobile application Sesamo. Sesamo passively tracks location in the background using variable frequency based on speed of movement, with a granularity of one second. Accordingly, the application has a low burden on battery and CPU usage of mobile devices. After installing the application, it required no further input from the participants. Data were processed by matching with street and transportation network where possible. Resulting data comprised over 800,000 geographic coordinate pairs.

Due to the variable frequency of recording, it is important to keep in mind that the data as analyzed are representative of participant location in terms of their *movements* rather than their *time expenditure.* That is, a larger number of data points in a specific location reflects not only more participants, but specifically more participants *on the move* in that area, rather than participants necessarily lingering or *spending more time* in that area.

#### 3.3.2. Self-Reported Emotion

We asked participants to report on their daily emotions using a modification of the Scale of Positive and Negative Experience SPANE, [47]. In general, self-response measures of emotion follow a ‘gold standard’ of presenting participants with a list of common-language emotion terms, such as “joyful” and “angry” and asking them to rate the extent to which they felt each emotion in the list over a given span of time. Responses usually load on positive and negative emotion factors that are internally consistent. Herewith, asking tourists to report per day each evening of their vacation is a well-tested approach [33,48]. Participants were asked to indicate the extent to which they felt each emotion over the course of the day on 5-point Likert-type scale from “Not at all” to “Extremely.”

While various emotion scales differ in the specific list of emotions presented, the SPANE is brief and well-balanced between positive and negative emotions. There are four general emotion terms in the SPANE which may not be strictly measuring emotion (“Good”, “Bad”, “Pleasant”, and “Unpleasant”) and were thus omitted. An emotion known to be important in tourism experiences, “positively surprised”, was added. We averaged the five positive and four negative emotion items together into positive and negative emotion indices. Positive emotion items were internally consistent (Cronbach alpha = 0.86; Revelle’s omega = 0.88). Negative emotion items were less internally consistent, in line with the very low variation normally present on negative emotion items in many tourism experiences (Cronbach alpha = 0.62; Revelle’s omega = 0.69). The low internal consistency in this case can specifically be attributed to the item “afraid”. We chose to retain this item as removing it improved the internal consistency only somewhat, and it is a conceptually important negative emotion.

#### 3.3.3. Experience Evaluation

We measured four dimensions of experience evaluation using single-item measures based on the Net Promotor Score item on intent to recommend [49]. Accordingly, we asked participants how likely they were to recommend the destination region (the province of Overijssel), their accommodation (the specific vacation park where they stayed), and the source of information to which they were assigned (passive app or conversational recommender), on a Likert-type scale ranging from 0 (Not at all likely) to 10 (Extremely likely). This intent to recommend item has been validated, and most widely applied, with this 0-to-10 response format.

While these item address participants’ evaluations of specific dimensions of their vacation, they do not cover the experience evaluation of the vacation as a goal. To that end, we used a single item with the same 0-to-10 Likert-type response scale, this time asking participants to give their vacation as a whole a grade, with 0 being the worst possible, and 10 being the best possible. This scale is familiar to Dutch participants as the grading scale used for school exams, and has been successfully validated in measuring experience evaluations in The Netherlands in previous research [35]. Furthermore, as overall grade and intent to recommend are adjacent in the questionnaire, using the same response scale for both prevented the cognitive burden of excess shifting between scale formats.

### 3.4. Analyses

We analyzed the data in five stages. First, we described the experience and evaluation variables for the sample as a whole. Then, we examined differences on experience and evaluation between the four conditions (popularity-driven/passive, policy-driven/passive, popularity-driven/conversational, policy-driven/conversational) using conditional group means and one-way analysis of variance. The third and fourth stage of data analysis aimed to assess if participants in different conditions visited different locations.

In the third stage, we processed GPS data by first eliminating any data points which were not between date of arrival and date of departure or located outside of Overijssel. To analyze spatial patterns of participant distribution we transformed the participant point locations captured by GPS to continuous density representation using bivariate kernel density estimates [50]. The kernel density maps were computed for participants grouped by recommender system conditions and popularity-driven and policy-driven conditions. We then mapped differences in kernel densities between recommender system conditions (passive vs. conversational) separately for popularity-driven and policy-driven conditions.

The last stage of data analysis involved modeling spatial movements of participants within spatial buffers generated around the attractions (20 m for point attractions, such as museums, restaurants, and monuments and 100 m for area-type attractions, such as parks or playgrounds) as a function of experimental condition. In other words, we modeled the odds that each datapoint collected came from a participant in one of the experimental conditions. Data were nested within participants, as both emotions and spatial behavior tend to be autocorrelated within participants. A multilevel logit model was used. We ran three models with different types of attractions as the outcomes: non-priority attractions, attractions which had popularity-driven priority, and attractions which had policy-driven priority. Finally, we also used these attraction movement variables as predictors of emotions to explore if participants enjoyed their vacation more on days when they engaged in more movements at a specific type of attraction.

The popularity-driven passive condition was the reference group for all analyses, meaning all other conditions were always compared to this one, because a popularity-driven passive map-based application is the current default option used by many DMOs, and is thus analogous to a control condition in the present study. For smaller DMOs which do not have their own app, printed materials and conversations in tourist information centers also tend to focus on maps with certain attractions highlighted. Thus, the coefficients of models explaining participant movements based on experimental condition express the differences in movements between participants in the popularity-driven passive condition and each other condition in turn.

## 4. Results

### 4.1. Descriptive Statistics

An initial group of 269 participants filled in the recruitment form and intake questionnaire. Random assignment led to a relatively even division across the four experimental groups (popularity-driven passive n = 71; policy-driven passive n = 65; popularity-driven conversational n = 61; policy-driven conversational n = 72). Of these, 268 filled in at least one daily questionnaire and 132 filled in a daily questionnaire on the last day of their vacation. The responses between the different questionnaires presented to each participant do not overlap fully. The sample as measured by the intake questionnaire was over three-quarters female (76%) with a mean age of 44 years (sd = 11 years). About two-thirds (65%) of participants possessed bachelor or higher degrees of education, while the remaining 35% had vocational degrees. A large majority went on vacation with either their partner (14%) or family (79%). The average age fits almost exactly with previous national research on visitors to Overijssel (M. Kompanje, personal communication, date). Participants had booked their stay for an average of 9.33 days (sd = 5.54).

Participants generally enjoyed their vacations, the destination, and the destination information source to which they were assigned. On average participants graded their vacation with a 7.77 (sd = 1.25) and were quite likely to recommend their accommodation (mean = 8.10, sd = 1.67) and Overijssel (mean = 8.38, sd = 1.15). They were also mildly positive about the information source to which they were assigned (mean = 6.08, sd = 2.71). Average daily positive emotions were approximately normally distributed, with a mean of 3.19 on a 5-point scale (sd = 0.58). Negative emotions were extremely positively skewed, as usual for tourism datasets, with very few participants reporting much of any negative emotion at all (mean = 1.31, sd = 0.28).

### 4.2. Differences between Groups

There were remarkably few differences in experiences and outcomes between groups. The groups were statistically similar in positive emotions on vacation (F = 1.028, *p* = 0.3812), negative emotions on vacation (F = 2.22, *p* = 0.0869), overall grade of vacation (F = 0.5418, *p* = 0.6546), intent to recommend Overijssel (F = 0.7538, *p* = 0.5222), and intent to recommend their accommodation (F = 0.6399, *p* = 0.5908). In other words, most group means were very close to one another, given the response scale (Figure 2a–e).

There were large differences between groups in evaluations of destination information sources. The passive app with either kind of tips earned about a 5 on an 11 point scale ranging from 0 to 10. The conversational recommender, on the other hand, earned a 6.7 (popularity-driven) to 7.3 (policy-driven) on the last day of vacation (F = 10.11, *p* < 0.001). Differences between tourists’ experiences aggregated by recommender system are summarized in Table 1 and Figure 2f.

### 4.3. Spatial Distribution between Groups

As a whole, participants moved around the most near their accommodations, but spread out over the entire province of Overijssel, including all its major cities and motorways as well as side roads. On a map showing locations visited by at least one participant from each group (Figure 3), it is evident that all groups were present on roads around the vacation parks, as well as between the vacation parks and Enschede. Meanwhile the most visited areas (highest kernel densities of data points) are concentrated around the accommodations and the historic Hanseatic cities of Deventer and Zwolle (Figure 4). Subsequent maps are based on data from the 6 accommodations that had at least 10 participants per experimental group. Statistical analyses are not limited to participants from these 6 accommodations, and instead use all available data.

Unlike differences in reported experience and evaluations, the differences between groups in terms of where they went were clear and striking. We created maps showing locations where only one of the four groups was present (Figure 5 and Figure 6). Patterns here are difficult to discern, but there are substantial segments of provincial roads around Zwolle, Kampen, Staphorst, Tubbergen, and Enschede that were only visited by a single group. This points to different groups aiming at different attractions. Furthermore, descriptive statistics show that the different groups covered different proportions of the geographic area of Overijssel. While popularity-driven passive and policy-driven conversational groups visited about two-thirds of visited areas (and thus, 16% of the total area of Overijssel), the policy-driven passive group visited only one-third (and thus only 9% of Overijssel; Table 2). The patterns in these data show that the policy of ‘spreading’ tourists over a larger geographical area can be driven by changing attraction information from the default, but only if recommended conversationally rather than passively. The same pattern is accentuated in urban areas, where participants in the conversational policy-driven condition exhibited by far the most coverage.

Maps of kernel density differences showed that policy-driven participants made more movements west of Ommen and in the municipalities of Rijssen-Holten and Hardenberg. Popularity-driven participants, on the other hand, moved around more just east of Ommen, just west of Almelo, and on the north side of Zwolle. (Figure 7).

Dividing popularity-driven and policy-driven participants into two separate maps, we examine differences between the recommender systems. Popularity-driven participants using the passive app were more present in Enschede, Raalte, and Staphorst, comprising 57% of the visited locations on this map, while they were more present around Deventer, Rijssen-Holten and south of Hardenberg if used the conversational recommender, comprising 43% of the visited locations on this map (Figure 8).

Policy-driven participants, on the other hand, showed almost the opposite pattern. They were present south of Hardenberg and around Tubbergen if using the passive app, comprising just 21% of the movement locations on this map, but more present around Raalte, Rijssen-Holten, and Ommen if using the conversational recommender, an overwhelming 79% of movement locations (Figure 9).

Statistical models assessing number of points at attractions of various types (non-priority, priority policy-driven, and priority popularity-driven) as a function of group confirms that tourists in different groups not only went to different places but went to the *locations prioritized by the information they received* from the experimental intervention. There were no differences between groups in movements at non-priority attractions. At priority popularity-driven attractions, there was no significant difference between movements of popularity-driven and policy-driven conversational recommender users, but passive app users who received policy-priority information were only 0.12 times as likely to be recorded at popularity-driven attractions as passive app users getting popularity-priority information. In other words, among participants in passive information conditions, those who got policy-driven information visited popularity-driven attractions 88% less.

At policy-driven attractions, participants getting policy-driven information were 1.8 (passive) to 2.0 (conversational) times as likely to be recorded at policy-driven attractions, as participants getting popularity-driven information by passive app. Interestingly, conversational recommender users getting popularity-driven information were *also* 1.5 times more likely to be present at *policy-driven* attractions (approaching significance at *p* = 0.06) compared to participants getting popularity-driven information by passive app. Further differences emerged when looking at movements at urban areas. While groups receiving information passively were equally likely to be recorded in urban areas, groups receiving information conversationally were only 0.47 as likely to be recorded in urban areas if the information was popularity-driven, and only 0.15 as likely to be recorded in urban areas if the information was policy-driven. In other words, participants in the policy-driven conversational condition moved around in urban areas 85% less than participants in the popularity-driven passive condition (Table 3).

### 4.4. Experience over Space

There were no significant effects of aggregated daily movements at different attraction types on daily positive emotions, within participants. There were modest connections between spatial behavior and negative emotions, however. Days when participants spent relatively more movements at non-priority attractions featured more negative emotions. In contrast, days with more movements at popularity-driven locations featured fewer negative emotions. Thus, it could be said that the most negative days had the most movement at non-priority attractions, while the least negative days had the most movement at popularity-driven attractions, on average within participants (Table 4). An important caveat for this finding is that in many tourism settings, whatever emotions tourists might actually experience, they report fairly low negative emotion, deflating any variation that models of this variable could potentially explain.

## 5. Discussion

### 5.1. Theoretical Implications

Our findings show that intervening on information changes tourists’ spatial behavior substantially, but has only limited implications for their experiences, extending existing knowledge in several ways. First, we assert that information provided to tourists firmly belongs in the theoretical discussion about causes and consequences of overtourism. While much has been written about the macro-level social forces that cause a destination to be crowded and bring its local residents into conflict with tourists [1,6], scholars have been reticent to suggest and especially test solutions. The present findings break this silence, clearly demonstrating that informing tourists about less-visited attractions increases their visitation to those attractions, while reducing their movements around more popular ones. There are also differences in which cities participants visited on a regional level, with some urban areas and access roads *only* visited by tourists with certain kinds of information. Furthermore, while participants approached with information conversationally moved less around urban areas, they also spread over a greater percentage of urban area if this information was adjusted with the policy goal of spreading them. Thus, we add the promising solution of information delivery to the evolving discussions around overtourism and urge scholars to further test and refine informational and other solutions to address this issue. The patterns in which urban areas are visited to what extent under different information strategies is particularly promising as cities in other contexts struggle with greater overtourism problems.

Second, we show that addressing crowding using an information intervention is moderately more effective with a conversational recommender, which is also much more positively evaluated, than with a passive map-based app. While positive experiences of personalized recommender systems in tourism were previously documented [19], our study is consistent with these findings, and contributes the distinction between effects of the conversational recommender on the experience as a whole, which was minimal, on spatial behavior, which was moderate, and on the experience of receiving information, which was strongly positive. These findings are also consistent with the apparent early promise of recommender systems [21], which was empirically validated as driven by personalization [20]. The hypothesis in [3] that destinations would benefit from personalizing information toward visitors is thus validated by our findings, especially in instances where DMOs wish to co-brand information sources such as recommenders. Furthermore, the advantages of a conversational system are accentuated when looking at urban areas, which were visited by fewer tourists receiving information conversationally, but with a wider spatial distribution over those urban areas. The use of an experimental design effectively frames these effects in comparison to contemporary default destination information practices.

Third, our study contributes evidence of modest variation in tourist experience over space on the temporal scale of differences between days over an entire vacation. Previous studies showed that single tourist experience episodes (e.g., city walks; guided tours) vary over urban space [32,44,45], whereas our findings uncovered rather little effect of a spatial predictor, attraction types, on the larger temporal scale of an entire vacation. The spatial scale in the present study was also larger; we tracked participants from city to city rather than within a single urban space.

Correspondingly, vacations were evaluated equally no matter of which attractions tourists were informed. Thus, our findings dispel the persistent myth in destination management that promoting only lesser-visited attractions will ‘ruin’ a vacation. We surmise that this may be due to novelty, the experience that vacation surroundings are new and different from everyday life. This explains a substantial proportion of the positive emotion boost people experience during vacationing [40], and may be similar regardless of whether attractions visited are popular or lesser-known. The relationships between information provision, spatial behavior across attraction types, and vacation experiences deserves more study. Especially promising is the possibility to combine physiological emotion sensing [44] with multi-day vacation recording, such as in the present study, using a complex methodological approach such as measurement-burst design [51].

### 5.2. Professional Implications

Our findings suggest that the current trend to shift destination management policy from attracting as many tourists as possible, to spreading tourists away from crowded attractions to less-popular ones, can be implemented by intervening on information delivery. Specifically, at least those tourists who are open to destination information can be led to greatly reduce their visits to crowded attractions while potentially almost doubling their visits to lesser-known places. Destination Management Organizations (DMOs) should therefore critically examine where tourists obtain information. Tourists respond to digital recommender systems, whether passive or active, by visiting different attraction locations. Herein the most advanced conversational systems are likely to be the most powerful, because they personalize information more effectively and are more positively evaluated. In our findings, a highly personalized recommender was evaluated much more passively than a default information source, a passive map-based app, and was also mildly more effective in redirecting tourist flows. Thus, we encourage municipalities, regional governments, and DMOs to actively communicate where they wish tourists to go, and deactivate promotion of locations where they would like to reduce crowding. In our findings, such an intervention was not associated with lower experience evaluation. An information intervention is likely to be effective within a passive recommender system. It may be more somewhat effective with a conversational one, as it was in our data. Furthermore, considering that many destinations now lend their brand to passive map-based apps, our findings would suggest that a more personalized and conversational channel for delivering information may reflect better on the destination brand. The extent to which information source evaluations reflect back on destination brands is an interesting and relevant question for future research.

In terms of the connection between location and experience, popularity-driven priority attractions performed better in our study, with slightly lower negative emotions, than policy-driven attractions. Attraction reputation is very dynamic and difficult to grasp reasons in contemporary social milieus, where experience quality is quickly communicated between tourists on review sites and social media, a process which could be further explored in future research. Based on the difference between popularity-driven and policy-driven attractions we found in terms of emotions, we urge DMOs to critically evaluate the quality of attractions they recommend, and to implement marketing policies while keeping tourists’ experiences in mind.

Finally, we recommend regional governments and other destination stakeholders to make decisions based on data, such as those in the present project. Collecting and analyzing these data requires investment in an appropriate data software infrastructure, but it is possible to start small, scale up to projects such as this one, and further yet to a reciprocal process which connects management decision cycles to recurring data collections on tourist experience and behavior. Certainly, *any* methodologically rigorous research on tourist behavior and experience is bound to lead to better destination management decisions than pure intuition.

### 5.3. Limitations and Future Directions

During the first wave of data collection (May 2021), museums and restaurants were closed due to COVID-19-related lockdowns. Many people were hoping for some perspective on the situation before booking a vacation. At the last minute, when they might still have booked, the weather turned out to be cold and very rainy. Thus, it was difficult to collect a large number of participants, and it was difficult to deliver information about attractions that they would actually be able to use. We initiated a second data wave of collection at the end of July and beginning of August, during much more favorable conditions. The situation in domestic tourism in The Netherlands is once again different, however, with all pandemic measures having been lifted at the time of this writing (early 2023), international visitation climbing toward pre-pandemic levels, and domestic tourism at vacation parks remaining popular. Thus, replicating the present study under current conditions could uncover different results, as tourists today face different levels of crowding than in 2021.

The accommodations at which we sampled do not comprise a probability sample of accommodation bookings or of vacation parks in Overijssel. More generally, one might ask how representative our sample was of tourists that marketers are hoping to reach with information. Not every tourist will seek out or pay attention to destination information from organizational channels. In that sense, it is likely that the same tourists which downloaded information for an experiment such as this one would also download information purely for use during their vacation. Thus, our participants maybe have been different in ways that made them more susceptible to change their behavior in response to the information provided. It also raises the question if a passive map-based app was an optimal control condition in our experiment. Certainly, more experimental conditions with different channels for the provision of information, as well as a condition with no information at all provided, could lead to deeper insights. In future research we advocate deriving sources of information to be tested from the bottom up, based on observations of or interviews about tourists’ behavior. This could increase the validity and usefulness of the conclusions.

Between-individual differences in response to the experimental intervention were controlled for by the random assignment used in our experimental design. However, we also analyzed the within-individual effect of location on experiences, which could be moderated by between-individual variables such as length of stay or travel motivations. A limitation is that we did not include these variables in our analysis. We also did not collect extensive demographic information, such as household composition or place of residence, which can lead to appreciable differences in how vacations are experienced; for example in how far away, new, or different the vacation setting feels. Controlling these variables or modeling these moderations in future research could lead more precise insights into the relationship between location and experience.

Analyzing a contextual variable as complicated as location also brings limitations. We chose to use a buffer approach to measure participants’ spatial behavior in relation to attraction locations, meaning when data points were within a certain minimum distance of an attraction, we counted them as being “at” that attraction, as a reasonably simple and accurate estimate. As with all measures of social behavior, it was imperfect, however. Every single data point within a buffer does not mean that the participants visited the attraction, just that they were nearby. Combining buffers with kernel densities or per-participant time data would address this issue. For area-type attractions the boundary polygons or trails would be needed to get more accurate estimate of the visitors.

## 6. Conclusions

The findings of the present study demonstrate that giving tourists policy-driven information, especially by conversational recommender system, has the potential to spread them away from heavily visited areas to a wider geographic area and specifically to attractions in less-visited areas. Their visitation of urban areas also shifts. Furthermore, giving tourists different information via different channels may not change their experience of their vacation, though it is likely to change their experience of receiving information. In sum, the findings show promise for informing tourists about lesser-known attractions to reduce crowding without degrading the quality of their experiences.

The findings situate actionable and affordable interventions in the often negative discussion about overtourism and increasing crowding in areas with vigorous tourist flows. We highlight the importance of attending to information that tourists use to make spatial behavior decisions at the destination. The present study shows that information interventions may carry the potential to improve quality of life for all destination stakeholders, including both tourists and local residents. The promise of intervening on information delivered to tourists is particularly substantial if the effects of these interventions are measured, and further decisions are made based on evidence.

## Figures and Tables

**Figure 1 ijerph-20-05441-f001:**
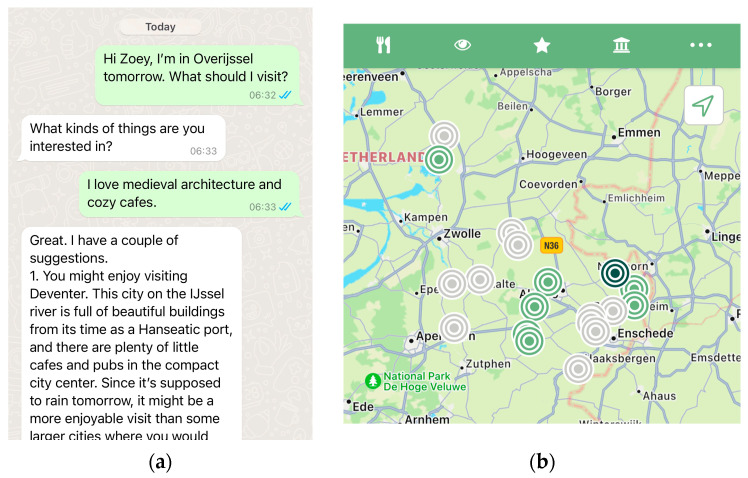
Screenshots of conversational (**a**) and passive (**b**) recommender destination information.

**Figure 2 ijerph-20-05441-f002:**
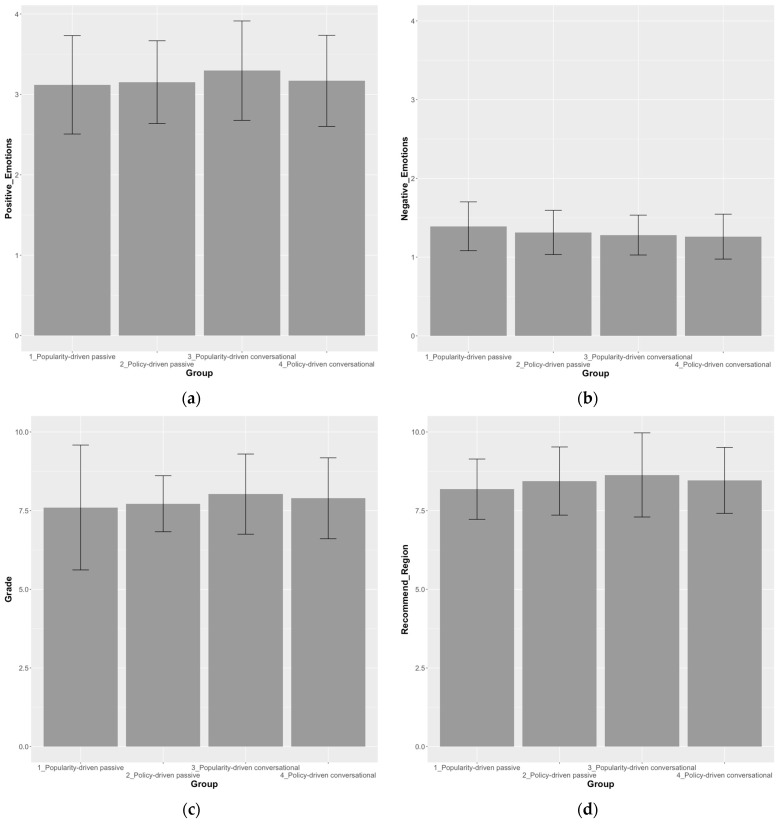
Self-reported differences between groups on (**a**) positive emotions, (**b**) negative emotions, (**c**) overall grade for vacation, (**d**) intent to recommend Overijssel, (**e**) intent to recommend accommodation, (**f**) intent to recommend the recommender system.

**Figure 3 ijerph-20-05441-f003:**
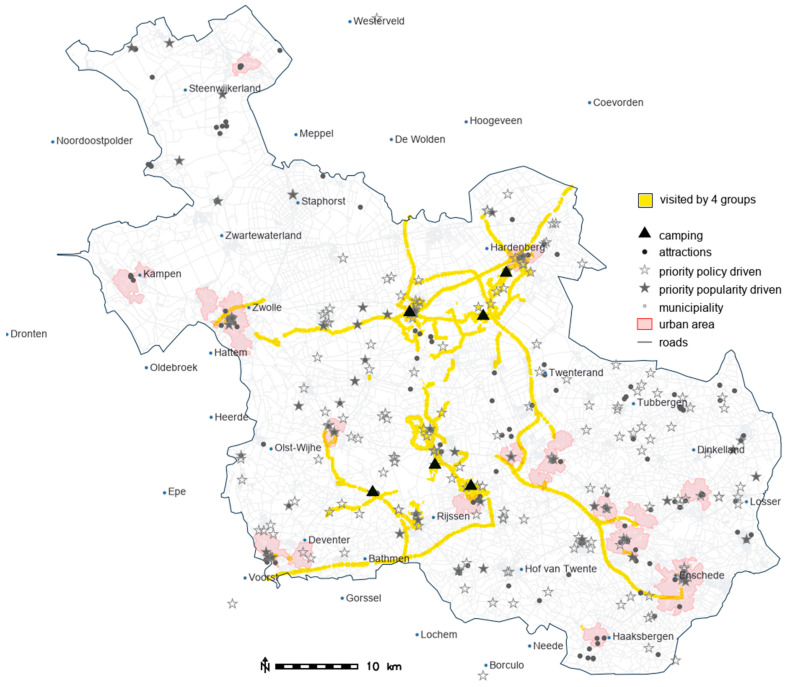
Locations visited by at least one participant from all four groups.

**Figure 4 ijerph-20-05441-f004:**
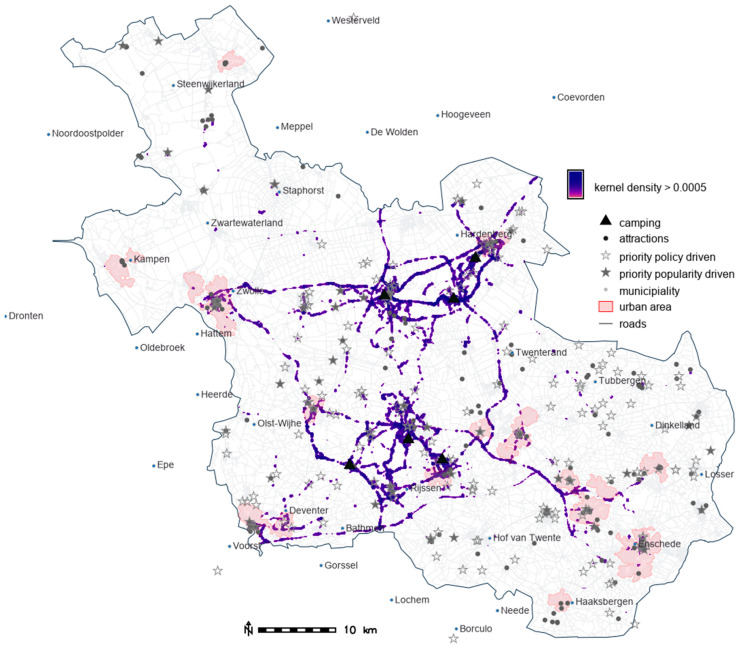
Most visited locations.

**Figure 5 ijerph-20-05441-f005:**
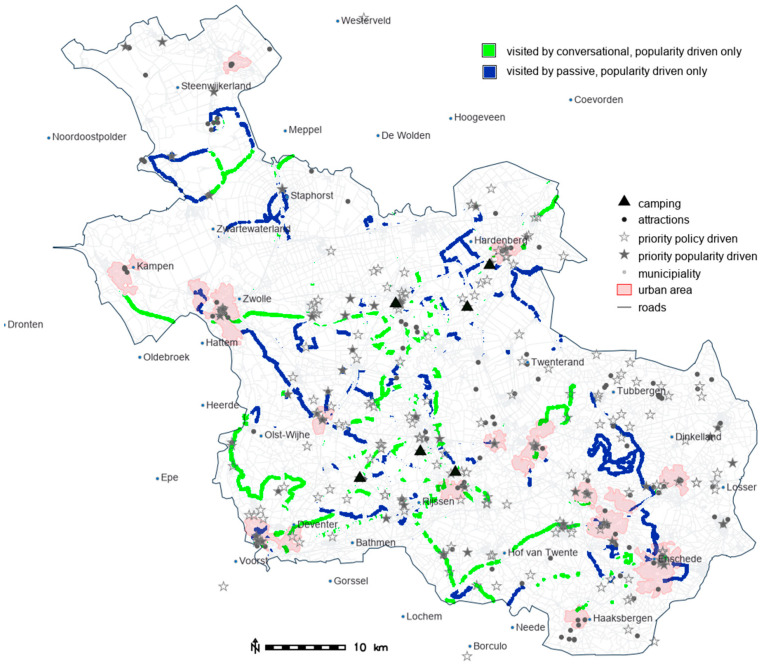
Locations visited exclusively by each of the 2 popularity-driven groups.

**Figure 6 ijerph-20-05441-f006:**
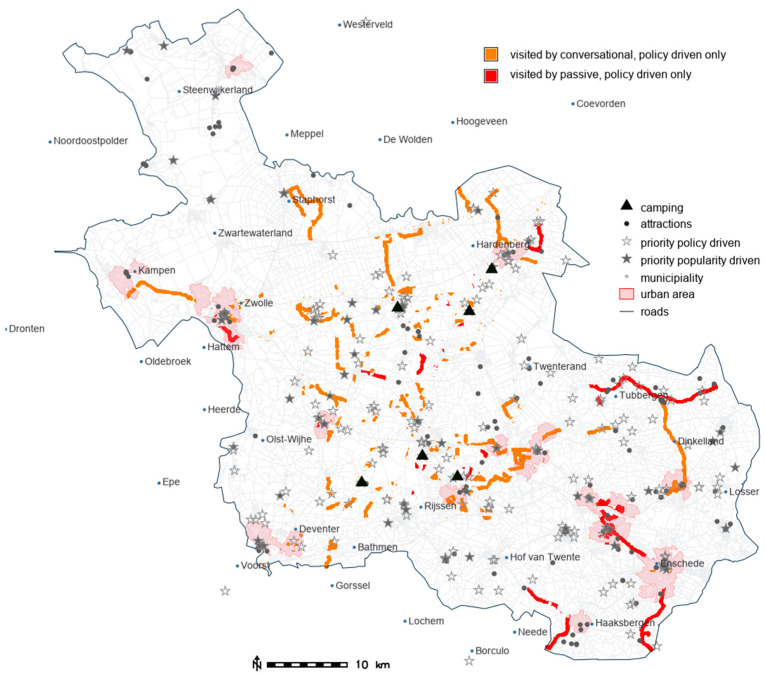
Locations visited exclusively by each of the 2 policy-driven groups.

**Figure 7 ijerph-20-05441-f007:**
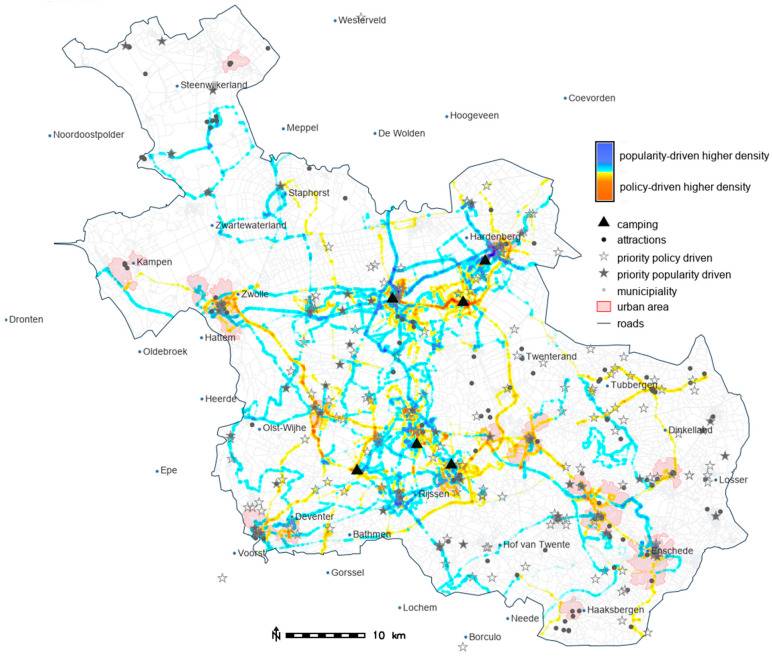
Difference between popularity-driven and policy-driven participants.

**Figure 8 ijerph-20-05441-f008:**
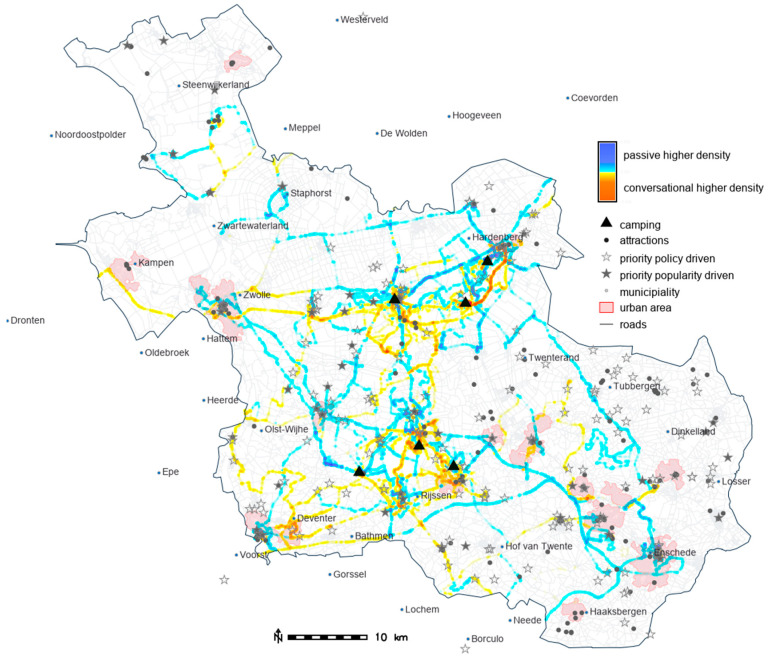
Difference between passive and conversational popularity-driven participants.

**Figure 9 ijerph-20-05441-f009:**
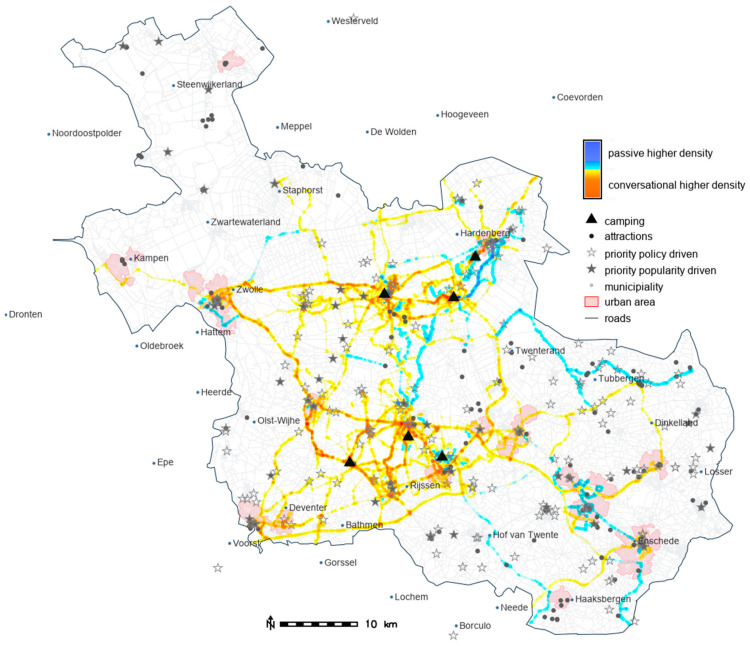
Difference between passive and conversational policy-driven participants.

**Table 1 ijerph-20-05441-t001:** Self-reported differences between groups including inferential comparisons to passive popularity-driven group.

		Experimental Group			
	Scale	Passive Popularity-Driven	Passive Policy-Driven	Conversational Popularity-Driven	Conversational Policy-Driven
Positive emotions on vacation	1–5	3.12	3.15	3.29	3.17
Negative emotions on vacation	1–5	1.39	1.31	**1.28 ***	**1.26 ***
Overall grade for vacation	0–10	7.60	7.72	8.02	7.89
Intent to recommend Overijssel	0–10	8.18	8.44	8.63	8.46
Intent to recommend accommodation	0–10	7.82	8.48	8.07	8.03
Intent to recommend the recommender system	0–10	4.45	5.16	**6.66 *****	**7.36 *****

Note: Significant differences compared to passive popularity-driven group marked as * *p* < 0.05, ** *p* < 0.01, *** *p* < 0.001.

**Table 2 ijerph-20-05441-t002:** Coverage of Overijssel and its urban areas by each group.

Group	% of Visited Area that Was Visited by This Group (within Urban Areas)	% of Visited Area that Was Visited Only by This Group (within Urban Areas)	% of Overijssel Visited by This Group (within Urban Areas)
Popularity-Driven Passive	66% (55%)	13% (9%)	17% (27%)
Policy-Driven Passive	36% (44%)	4% (10%)	9% (22%)
Popularity-Driven Conversational	56% (44%)	10% (6%)	14% (22%)
Policy-Driven Conversational	61% (66%)	10% (15%)	15% (32%)

**Table 3 ijerph-20-05441-t003:** Effect of recommender systems and destination information on spatial movement, expressed as odds ratios in reference to popularity-driven passive group.

Popularity-Driven Attractions			
	Group	Odds Ratio	Standard Error
	(Intercept)	0.00	0.46
	**Policy-driven passive**	**0.12 ****	**0.75**
	Popularity-driven conversational	0.40	0.91
	Policy-driven conversational	0.30	0.75
Policy-driven attractions			
	(Intercept)	0.01	0.13
	**Policy-driven passive**	**1.80 ***	**0.20**
	Popularity-driven conversational	1.51	0.22
	**Policy-driven conversational**	**2.02 *****	**0.24**
Non-priority attractions			
	(Intercept)	0.00	0.23
	Policy-driven passive	0.89	0.36
	Popularity-driven conversational	1.18	0.34
	Policy-driven conversational	1.26	0.33
Urban areas			
	(Intercept)	0.03	0.11
	Policy-driven passive	1.02	0.25
	Popularity-driven conversational	**0.47 *****	**0.14**
	Policy-driven conversational	**0.15 *****	**0.23**

Note: Popularity-driven attraction model AIC = 10,848.9; BIC = 10,904.4; Policy-driven attraction model AIC = 103,548.9; BIC = 103,604.4; Non-priority attraction model AIC = 41,765.2; BIC = 41,820.7; Urban area model AIC = 272,864.75; BIC = 272,920.26; * *p* < 0.05, ** *p* < 0.01, *** *p* < 0.001.

**Table 4 ijerph-20-05441-t004:** Effects of movement at attractions compared to movement at non-attraction locations on daily emotions.

Outcome Variable	Predictor	Coefficient	Standard Error
Positive emotions			
	(Intercept)	3.278	0.046
	Movement at policy-driven attractions	−0.006	0.004
	Movement at popularity-driven attractions	0.021	0.015
	Movement at non-priority attractions	−0.009	0.007
Negative emotions			
	(Intercept)	1.128	0.009
	Movement at policy-driven attractions	0.000	0.001
	**Movement at popularity-driven attractions**	**−0.008 ***	**0.003**
	**Movement at non-priority attractions**	**0.008 *****	**0.002**

Note: Positive emotion model AIC = 574,279.4; BIC = 574,346.0; Negative emotion model AIC = −916,767.6; BIC = −916,701.0; * *p* < 0.05, ** *p* < 0.01, *** *p* < 0.001.

## Data Availability

The data used in this study are unavailable for public use due to the privacy sensitivity of location data that may inadvertently occasionally include participants’ travel from their home to their vacation location.

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
