# Peer review of "Tell Me Where to Go: An Experiment in Spreading Visitor Flows in The Netherlands"

_ijerph, 2023, doi:10.3390/ijerph20085441_

Round 1

Reviewer 1 Report

This will make a nice contribution to the literature. I do have some comment to consider in your revision:

1) line 16. "tourists" is not spelled correctly.

2) lines 208-220. Just because something has not been examined does not necessarily provide justification for undertaking of a study addressing such a gap. I concede it does it qualitative designs often; not so much in quantitative experimental designs. Work on conveying to the reader how, by carrying out this study, the study aims to make practical contributions and what those may be. 

3) line 242. What makes it 'true?' Be careful here using such a term. Also, in the same line, did random selection to be part of the study happen prior to random assignment?

4) section 2.1. For what variables did you control?

5) lines 255-287. Some tourism context for Overijssel would be most helpful here. What is the number of visitors in a given year? How many attractions are there? What are they? How many residents live there? This will help the reader gain a better sense of "overtourism" (as you allude to in section 1.1.1) that is occuring in the province.

6) section 2.2. How many participants were part of your study? What incentive did individuals have for participating in your study?

7) section 2.3.2. I think the section title/heading here should reflect the fact that you were addressing self-reported or perceived emotional states, no emotions in general.

8) section 2.3.3. The scale 0-10 is arguably not commonly used within many tourism studies. Furthermore, your justification for using it re: the acceptance of such a scale based on potential acceptance rooted in early public education seems rather weak.

9) section 2.5. This should be moved to the very end of your paper and include future research directions in response to such shortcomings.

10) section 3.2. First paragraph in section. How do you explain this lack of significant differences.

11) 

Reviewer 2 Report

Directing tourists to less-crowded location is a very interesting topic. Below are the suggestions for authors to improve their paper.

Introduction

Literature review should be separated from introduction. Move literature review to chapter 2 and clearly state research gap in introduction.

Literature review

Overtourism is a research background, there is no need to have too much discussion in literature review. Strengthen the literature on the influence of information on tourism flow, tourists’ experience and behavioral intention. The relationship among information, conversational recommender systems, tourism flow, tourism experience should be more tighten up.

Methodology

Justify the reason of using availability sampling. It seems that the respondents need a high degree of cooperation to complete the experiment. Are there any incentives to improve their degree of cooperation?

Are there any criteria for recruiting samplings? What if they have been to the destination many times? Does this experience affect the result? How about the confounding effects control?

Regards to the measure of experience evaluation, the authors measure tourists’ intention to recommend rather than their travel experience.

Discussion and implication

Are there any papers that discuss how information and conversational recommender systems influence tourist emotion and experience? Are the findings consistent with previous research?

This part should contain discussion, theoretical implications, professional implications, and limitation.

The practical implication should be more based on the research result.

All the best to the authors.

Reviewer 3 Report

The manuscript is quite interesting and show the potential of using intervening method to relive the overtourism issue. Several comments / questions are listed below.

1. For the study design, it would be better to also include the “control” group, which includes the participants that not receiving any information.

2. For the current participants, it would be better to get more details on participant’s demographic information. For example, how many days the tourists will stay in Overijssel, the short-stay and long-stay participants will have different movements; where are the participants from, are they form the local people or from outside Overijssel province, are they from urban area or non-urban area. If many participants are local, their movements may not reflect their travel behavior, but their daily activity behavior.

Minor suggestions

1. In Line 16, “tourist s” should be “tourists”

Round 2

Reviewer 2 Report

Thanks to the authors efforts, the quality of the revised version has improved significantly.

There are some minor suggestions.

The study aims to compare the effectiveness of conversational and passive recommender systems. The definition of these two types of systems should be provided in the introduction. I noticed that the authors mentioned conversational recommender system (page 2, line 58), but not passive recommender system. In addition, the significance of the study should be mentioned in Introduction.

Whether the two information systems (conversational vs. passive) selected by the authors are consistent in the quantity and quality of information provided, except for the conversation function. Because the amount and quality of information can affect the emotion and experience of tourists. A screenshot of the systems will help readers better understand the differences between the two systems.

Good luck.
